# Is refugee experience in childhood a risk for poorer health in adulthood?—A Swedish national survey study

Erica Mattelin[1,2,3]*, Amal R. Khanolkar[4,5], Laura Korhonen[1,3,6], Jill. W. Åhs[7,8], Frida Fröberg[1,9]

**1** Department of Biomedical and Clinical Sciences, Barnafrid, Swedish National Center on Violence Against Children, Linköping University, Linköping, Sweden, **2** Save the Children, Stockholm, Sweden, **3** Department of Biomedical and Clinical Sciences, Center for Social and Affective Neuroscience, Linköping University, Linköping, Sweden, **4** Department of Population Health Sciences, School of Life Course and Population Studies, King's College London, London, United Kingdom, **5** Department of Global Public Health, Karolinska Institutet, Stockholm, Sweden, **6** Department of Child and Adolescent Psychiatry and Department of Biomedical and Clinical Sciences, Linköping University, Linköping, Sweden, **7** Department of Neurobiology, Care Sciences and Society, Division of Nursing, Karolinska Institute, Huddinge, Sweden, **8** Department of Health Sciences, Swedish Red Cross University, Huddinge, Sweden, **9** Institute for Globally Distributed Open Research and Education (IGDORE), Gothenburg, Sweden

* erica.mattelin@liu.se

**Data Availability Statement:** Data used in the present study cannot be shared. However, data from the The Swedish National Public Health Survey can be accessed with the necessary permissions from the Swedish Public Health

## Abstract

Studies on experiences of migration in childhood and subsequent health in adulthood report conflicting results. While there is limited research on the long-term health outcomes of refugee children as they transition into adulthood, it is often observed that refugee children experience adverse health conditions upon their arrival in the host country. We examined whether adults with a childhood refugee experience were more likely to have poorer mental health, general health, and risk-behaviours compared to non-refugee migrants and Swedish-born peers We included a nationally representative sample of 18-64-year-olds who answered the Swedish National Public Health survey in 2018 or 2020. Using official register data, we categorized individuals as: 1) refugees in childhood (<age 18 years); 2) migrants in childhood; or 3) Swedish-born. Associations between childhood status (refugee, migrant, or Swedish-born) and all outcomes in adulthood were analyzed using logistic regression. The final model was adjusted for age, sexual- and gender-minority (SGM) identity, and stratified by sex. We also analysed the above and all outcomes stratified by age (18–25, 26–64), adjusted for sex and SGM-identity. We found that a childhood refugee experience was not associated with worse self-rated general or mental health, or more risk behaviours in adulthood, compared to non-refugee migrants or Swedish-born individuals. Additionally, adults with a childhood refugee experience had lower odds of at-risk alcohol use and substance use than Swedish-born peers. In general, a childhood refugee experience was not associated with worse self-rated health or risk behaviours in adulthood when considering age and sexual- and gender-minority status.

Agency which could be contacted at
folkhalsoenkaten@folkhalsomyndigheten.se.

**Funding:** The authors received no specific funding
for this work.

**Competing interests:** The authors have declared
that no competing interests exist.

## Introduction

Childhood is a sensitive period of development that can be negatively impacted by environmental factors with long-lasting consequences on health across the life course [1]. For example, exposure to adversities such as violence or neglect [2] are associated with an increased risk of poorer physical (e.g., obesity and cancer) and mental health (e.g., depression and posttraumatic stress disorder [PTSD]), risk behaviours (e.g., at-risk alcohol and drug misuse), social problems (e.g., lower educational achievement) in adulthood and lower life expectancy [2–5].

Refugee children face an elevated risk of exposure to different adversities, including various types of violence, financial hardship, and discrimination, before, during, and after the process of migration [6]. Many refugee children also experience mental health issues during the first few years after arrival in a host country. Refugee children have a higher prevalence of PTSD, depression, and anxiety disorders shortly after resettlement in a host country than native-born peers [7]. However, the long-term consequences of childhood adversities on health for these individuals who were refugees in childhood are less known as studies have mainly focused on non-refugee migrants [8]. Longitudinal studies examining exposure to migration-related adversities in childhood and subsequent mental health in adulthood show contradictory results. Some North American studies show a higher risk of psychological distress in adulthood for migrant children than non-migrants [9,10], while European studies report a lower risk [11,12]. A recent Swedish study showed a health advantage in adulthood for those who migrated <25 years of age compared to those who migrated at older ages [13].

The existing studies on health in refugee individuals show mixed results and have only focused on healthcare utilization. In a Danish study, former child refugees and non-refugee migrants had a lower risk of high-care utilization in young adulthood for affective disorders but higher odds for psychotic and neurotic disorders in young adulthood compared to their native-born peers [14]. A study examining the utilization of psychiatric care among young adult refugees in Sweden, both unaccompanied and accompanied, revealed that refugee individuals had a higher likelihood of receiving in-patient care and being subject to compulsory admission to a psychiatric hospital compared to the Swedish population, but numbers between refugee and Swedish populations were similar for outpatient care [15]. Another study focusing on young adult refugees in Nordic countries found that refugee females had a lower risk of in-patient care in Sweden than Swedish-born, whilst refugee men had a higher risk of in-patient care [16]. Other studies examining healthcare utilization for alcohol and substance use disorders among young adult refugees have reported a lower risk of in-patient care for alcohol-related disorders [17], but a higher risk of in-patient care for substance use [18]. However, corresponding studies for self-assessed health are lacking. There is a need for general-population-based studies to complement current evidence on healthcare utilization. The body of research at present relies on clinicians' assessments of a selective population actively seeking healthcare rather than the general population. Further, all these previous studies have focused on health in young adulthood. Knowledge of the long-term health of being a refugee in childhood is particularly important for Swedish public health policy as Sweden has historically maintained an inclusive migration policy, dating back to the 1970s when migration to the country started to increase [19]. Sweden has for example been rated high on the Migration Integration Policy Index (MIPEX) since 2007 and was rated top 10 at the latest rating in 2019 [20]. To nuance this picture there are several areas to consider in the context of current and historical migration policy in Sweden. In recent years, this applies to, among other things, access to health and medical care for refugee children [21], housing situations [22], lack of access to non-emergency care for adults, and ethnic discrimination in the labor market [23]. However, the high influx of asylum seekers during 2014–2015 led to changes in Sweden's migration policy and to

the implementation of restrictions on migration [24]. During the years with the inclusive migration policy, Sweden received one of the highest numbers of asylum-seeking children in the previous decade (244 857 individuals aged 0–17 years between 2012–2022) [25]. Between 2000 and 2021, a significant portion (45%) of all (children and adults) asylum-seekers in Sweden originated from Syria, Afghanistan, Iraq, and Somalia. Additionally, during the same period, Sweden granted 1,629,895 resident permits to migrant individuals for reasons unrelated to seeking asylum-permit such as work-permits [26].

To examine whether a refugee experience in childhood (<18 years of age) is associated with poor health and risk behaviours in adulthood (18–64 years), we analysed the association between refugee status in childhood (<18 years of age) and self-rated mental and general health and risk behaviours in adulthood (18–64 years) compared to non-asylum-seeking migrants and Swedish-born, controlling for age and sexual- and gender minority status, in the representative Swedish National Public Health Survey. We also examined if these associations differed by age and sex.

## Methods

### Ethics statement

The Swedish National Public Health Survey is conducted with ethics approval from the Swedish National Board of Health and Welfare (Dnr 920031208). All participants were given written information about the study and gave consent by participating in the survey. The present study received ethics approval from the National Ethics Review Board (Dnr 2020–02847) and has been performed in accordance with the ethical standards laid down in the 1964 Declaration of Helsinki and its later amendments.

### Study population and design

We retrieved data from the Swedish National Public Health Survey conducted by the Swedish Public Health Agency and Statistics Sweden. This is a postal/online survey sent out to a random sample drawn from a population frame of 16–84-year-old residents in Sweden, according to the Register of the Total Population [27,28]. The survey collects information on various health issues and sociodemographic data and is linked to official register data, such as demographic and socioeconomic registers, including STATIV, a longitudinal database for integration and migration held by Statistics Sweden. For this study, we combined the most recently conducted surveys in 2018 and 2020, as they provide information on each participant's country of origin. With participation rates of 42.1% in 2018 and 42.3% in 2020, our eligible sample consisted of 168,952 residents in Sweden. The survey was distributed in Swedish, English, and Finnish. The authors did not have access to information that could identify individual participants during or after data collection. A non-response analysis and calibration report have been conducted by Statistics Sweden [29]. As missing data was <5% for all variables, all analyses were run with complete case data.

### Variables

**Exposure of interest.** Refugee experience in childhood. In this study, all individuals were categorised into one of these three groups:

1. Those with refugee experience in childhood (0–17 years)

2. Those who migrated in childhood for reasons not associated with refugee status or asylum-seeking (0–17 years)

3. Those born in Sweden

**Assigning status of refugee and asylum experience in childhood.**  Information on being a refugee in childhood was obtained from the longitudinal integration database STATIV. Individuals with refugee status, subsidiary protection status, status as otherwise in need of protection, and quota refugees as per the Swedish Board of Migration's grounds for granting asylum in childhood were considered to have been refugees in childhood. This definition has been previously used in other studies [30]. For those who lacked information in STATIV, we defined refugee experience in childhood using data from the Total Population Register (TRP) on key variables like year of birth, year of immigration, and country of birth.

**Assigning status of migrant experience in childhood.**  Individuals who migrated in childhood for reasons not associated with being a refugee or asylum seeker were determined using data from TRP (year of birth, year of immigration, and country of birth). Assigning either refugee or migrant status in childhood was done in consultation with the Swedish Board of Migration.

**Assigning status of individuals born in Sweden.**  The status of individuals born in Sweden was defined by data from TRP (year- and country of birth). For a more detailed description of the assignment of status, see Table B in S1 Text.

Those who arrived in Sweden as migrants and refugees after 17 years of age, those who responded to the survey before the age of 18, and all individuals 65 years or older were excluded.

## Outcomes

**Mental ill-health.**  Due to a change in survey methodology, *psychological distress* in the previous two weeks was assessed using the short versions of the General Health Questionnaire (GHQ-5; [31]) in 2018 and the Kessler-6 [32] in 2020. To obtain a measure of psychological distress for GHQ-5, a sum index is calculated based on the first five questions. Participants with a value of 1 are defined as having reduced mental well-being. For 2020, we defined psychological distress based on the Kessler-6 scale, with six questions assessing non-specific psychological distress in the past four weeks with a sum score of 0–24. Scores between 13–24 indicate severe psychological distress. The GHQ-12 and Kessler-6 variables were merged into a single dichotomous variable.

Other indicators of mental ill-health included lifetime suicidal ideation ("Have you ever been in a situation where you seriously considered taking your own life?") and attempted suicide ("Have you ever attempted to take your own life?"). Both variables were dichotomized as No vs. Yes. (Table A in S1 Text).

**General health.**  Information about general health was assessed by the question, "How would you rate your general health?" with no time frame indicated. Respondents could choose from five options: very good, good, moderate, bad, and very bad. The five categories were merged into a dichotomous variable with two categories: 1. "Very good, good or moderate general health" and 2. "bad or very bad health".

**Alcohol- and substance use.**  The AUDIT-C [33] consists of three questions that assess an individual's alcohol consumption in the prior year. Total scores (ranging from 0 to 12) were calculated and dichotomised to distinguish individuals with and without at-risk alcohol use (scores >5 in females and >6 in males) as recommended by The Swedish Public Health Agency [27]. Two questions about illicit drug use defined *substance use* at any time in life with a three-response alternative; "never/more than a year ago/within the past year." Responders who indicated drug use within the last year were coded as *substance users*.

The outcomes were measured in adulthood, defined as ages 18–64.

### Potential confounders

**Sociodemographic information.**    Information on *officially registered sex (*female/male)
and *age* was obtained from TRP. The variable *sexual- or current gender identity (SGM)* (hetero-
sexual/sexual- and gender minority) was constructed using combined information from two
survey questions: how the respondent defined their sexual identity (heterosexual/ bisexual/
homosexual/other); and gender identity (cisgender/transgender).

**Analyses.**    First, we calculated proportions of sociodemographic characteristics, and men-
tal and general health across the three categories of the refugee/migrant/Swedish-born indica-
tor variable (Tables 1 and 2). To examine whether any differences in GHQ-5 and Kessler-6
influenced our analyses, separate analyses were performed for scores of these measures and
associated outcomes, which showed similar results.

Associations between the refugee or migrant indicator and all outcomes were analysed
using multiple logistic regression with those born in Sweden as the reference category. Regres-
sion models were first run using the full sample. Models were also run stratified by current life
stage; young adulthood (18–25 years), and later adulthood (26–64 years) as assessed in 2018 or
2020. The full-sample models were adjusted sequentially for confounders as follows

Model 1 was adjusted for age. Model 2 was adjusted for SGM-identity because there is a known
association between sexual/gender identity, exposure to violence, and poor health outcomes [34–
36], and identifying as a sexual- and gender minority is also grounds for asylum in some instances.
Model 3 was mutually adjusted for all confounders. Regression models were run assuming that the
exposure (childhood experience of being a refugee, migrant, or born in Sweden) preceded the out-
comes of interest. Models stratified by age groups were only adjusted for SGM-identity

We used calibrated population weights provided by Statistics Sweden in analyses. These
weights calibrate the sample to the known population, take into account the sampling design,
adjust for non-response, and use auxiliary information from official registers to achieve more
accurate estimates [36,37]. In this study, the weights accounted for the fact that we pooled the
2018 and 2020 samples. We used the Complex Samples add-on module to SPSS that can handle
complex samples and weighting and estimated variance with Taylor Series Linearization. Results
were considered statistically significant if the 95% CI did not include 1 (or zero), as appropriate.

## Results

The final study sample consisted of 89,416 individuals, including 1,189 individuals with refu-
gee experience in childhood 2,736 with migrant experience in childhood, and 85,491 Swedish-
born individuals. The average number of years between arrival in Sweden and follow-up was
15.99 for refugees and 34.71 years for migrants. Descriptive statistics are presented in Table 1
and Table C in S1 Text.

### The association between refugee or migrant experience in childhood and self-reported mental and general health in adulthood (18 to 64 years)

Overall, we found that having a migrant experience in childhood was associated with worse
mental and general health in adulthood, with some differences between females and males.

For females, having migrated to Sweden in childhood was associated with psychological dis-
tress in adulthood. In total, 21.4% of females with a refugee experience and 20.7% with a
migrant experience in childhood reported psychological distress, compared to 14.8% of
females born in Sweden (Table 2). This association remained when adjusted for potential con-
founders (age and sexual- and gender minority status) for females with a migrant experience
(Table 3 and Fig 1).

**Table 1. Proportions of potential confounders (demographic and socioeconomic characteristics) among those with a refugee experience in childhood, those with a migrant experience in childhood, or Swedish-born. in proportions between categorical variables were examined with Pearson´s chi-square test.** The numbers are unweighted counts (n), weighted proportions (%), and p-values.

| | Females N = 49,808 | | | | | | | Males N = 39,608 | | | | | | |
| | Refugee experience in childhood N = 587 | | Migrant experience in childhood N = 1568 | | Swedish-born N = 47653 | | p-value | Refugee experience in childhood = 602 | | Migrant experience in childhood = 1168 | | Swedish-born = 37838 | | p-value |
|---|---|---|---|---|---|---|---|---|---|---|---|---|---|---|
| **Age in 2018/2020** | N | % | N | % | N | % | .000 | N | % | N | % | N | % | .000 |
| 18–25 | 296 | 55.1 | 358 | 29.6 | 6285 | 17.7 | | 321 | 58.2 | 243 | 24.6 | 4627 | 17.3 | |
| 26–64 | 291 | 44.9 | 1210 | 70.4 | 41368 | 82.3 | | 281 | 41.8 | 925 | 75.4 | 33211 | 82.7 | |
| **Sexual identity** | | | | | | | .000 | | | | | | | .000 |
| Heterosexual | 453 | 83.9 | 1346 | 85.8 | 43809 | 91.4 | | 457 | 83.6 | 1026 | 87.8 | 35634 | 94.6 | |
| Sexual or gender minority | 85 | 16.1 | 179 | 14.2 | 3207 | 8.6 | | 87 | 16.4 | 105 | 12.2 | 1787 | 5.4 | |
| **Level of education attained (at time of survey)** [a] | | | | | | | .000 | | | | | | | .000 |
| Attended any college | 156 | 25.9 | 505 | 20.5 | 20410 | 36.9 | | 115 | 18.3 | 279 | 25.9 | 11363 | 27.5 | |
| Attended any high school | 93 | 13.9 | 507 | 31.5 | 17093 | 37.3 | | 103 | 17.9 | 486 | 39.4 | 18072 | 46.1 | |
| Elementary or lower | 39 | 8.0 | 196 | 10.1 | 3788 | 8.2 | | 61 | 12.5 | 156 | 13.4 | 3697 | 9.2 | |
| Not calculated | 237 | 52.2 | 322 | 27.9 | 6792 | 17.6 | | 286 | 51.2 | 221 | 21.4 | 5012 | 17.2 | |

[a] Low (No education or Middle school or lower; Average (2 years of upper secondary school or high school or 3–4 years of upper secondary school or high school) 3: Some higher education/ University or college, less than 3 years/ University or college, 3 years or more. Those under 25 were considered too young to have achieved their highest level of education and were labeled Not calculated.

**Table 2. Past-year prevalence of being exposed to physical violence, threats, discrimination, and any kind of violence, general and mental health and risk-behaviours among 89,416 individuals who answered the Swedish Public Health Survey in 2018 and 2020.** Differences in proportions between categorical variables examined using Pearson´s chi-square. The numbers are unweighted counts (n), weighted proportions (%) and p-values.

| | Females N = 49,808 | | | | | | | Males N = 39,608 | | | | | | |
| | Refugee experience in childhood N = 587 | | Migrant experience in childhood N = 1568 | | Swedish-born N = 47653 | | | Refugee experience in childhood = 602 | | Migrant experience in childhood = 1168 | | Swedish-born = 37838 | | p-value |
| | N | % | N | % | N | % | p-value | N | % | N | % | N | % | p-value |
|---|---|---|---|---|---|---|---|---|---|---|---|---|---|---|
| **General health** | | | | | | | .007 | | | | | | | ns |
| Bad/very bad | 34 | 5.7% | 122 | 6.9% | 2793 | 5.9% | | 24 | 2.4% | 64 | 7.0% | 1639 | 4.4% | |
| **Psychological distress** | | | | | | | <.001 | | | | | | | <.001 |
| Psychological distress | 128 | 21.4% | 320 | 20.7% | 7491 | 14.8% | | 93 | 14.7% | 160 | 13.3% | 4134 | 10.7% | |
| **Suicide ideation** | | | | | | | <.001 | | | | | | | <.001 |
| Yes | 89 | 13.4% | 344 | 21.6% | 7255 | 17.7% | | 60 | 10.5% | 185 | 18.4% | 4406 | 13.0% | |
| **Suicide attempts** | | | | | | | <.001 | | | | | | | <.001 |
| Yes | 40 | 6.6% | 125 | 7.8% | 2214 | 5.5% | | 21 | 3.6% | 69 | 7.1% | 1109 | 3.1% | |
| **At-risk alcohol use** | | | | | | | <.001 | | | | | | | <.001 |
| Yes | 26 | 3.3% | 223 | 12.3% | 7309 | 16.1% | | 34 | 5.7% | 229 | 17.9% | 8677 | 24.0% | |
| **Substance use[2]** | | | | | | | <.001 | | | | | | | ns |
| Yes | 18 | 2.9% | 59 | 5.2% | 922 | 3.0% | | 30 | 3.9% | 47 | 4.2% | 1542 | 5.9% | |

**Table 3. Associations between childhood refugee experience and migrant childhood experience and mental health in 89,416 individuals aged 18–64 from the Swedish National Public Health Survey.** Estimates are from multivariable logistics regression models (models adjusted for age and sexual- and gender minority status). Stratified by sex.

| | Males | | | Females | | |
|---|---|---|---|---|---|---|
| | Psychological distress | Suicide thoughts | Suicide attempts | Psychological distress | Suicide thoughts | Suicide attempts |
| | OR (95% CI) | OR (95% CI) | OR (95% CI) | OR (95% CI) | OR (95% CI) | OR (95% CI) |
| All | | | | | | |
| Swedish-born | Ref (1) | Ref (1) | Ref (1) | Ref (1) | Ref (1) | Ref (1) |
| Refugee experience in childhood | 0.99 (0.66–1.49) | **0.49 (0.31–0.80)** | 0.80 (0.39–1.65) | 1.08 (0.75–1.53) | **0.47 (0.30–0.72)** | 0.94 (0.51–1.74) |
| Migrant experience in childhood | 1.16 (0.85–1.59) | 1.31 (0.96–1.78) | **1.97 (1.23–3.16)** | **1.30 (1.04–1.63)** | 1.11 (0.89–1.39) | 1.24 (0.88–1.76) |
| | Males | | | Females | | |
| | At- risk alcohol use | Substance use | General health | At-risk alcohol use | Substance use | General health |
| | OR (95% CI) | OR (95% CI) | OR (95% CI) | OR (95% CI) | OR (95% CI) | OR (95% CI) |
| All | | | | | | |
| Swedish-born | Ref (1) | Ref (1) | Ref (1) | Ref (1) | Ref (1) | Ref (1) |
| Refugee experience in childhood | **0.19 (0.11–0.34)** | **0.31 (0.14–0.65)** | 0.61 (0.29–1.29) | **0.15 (0.08–0.29)** | 0.47 (0.22–1.03) | 1.15 (0.62–2.12) |
| Migrant experience in childhood | **0.68 (0.52–0.88)** | 0.57 (0.32–1.02) | **1.62 (1.04–1.54)** | **0.71 (0.54–0.92)** | 1.31 (0.82–2.10) | 1.21 (0.84–1.72) |

Regarding suicidal ideation and attempts, both females and males who migrated to Sweden as children (but not refugees) reported a higher prevalence, compared to Swedish-born females and males (Table 2). However, in logistic regression after adjustment for confounders, this association was only significant in men who had nearly twice the odds for suicide attempts (aOR 1.97, 1.23–3.16) compared to men born in Sweden (Table 3 and Fig 1).

The association between refugee/ migrant status in childhood and general health in adulthood followed a similar pattern. As seen in Table 2, both females and males who were migrants as children reported a higher prevalence of poor general health in adulthood than refugee and Swedish-born females and males. However, in adjusted analyses this association only remains for males with a migrant childhood (Table 3 and Fig 1).

## The association between refugee or migrant experience in childhood and self-reported alcohol and substance use at 18 to 64 years

Individuals with refugee (5.7%) or migrant (17.9%) experience in childhood were less likely to report at-risk alcohol use in adulthood compared to Swedish-born peers (24%, Table 2).

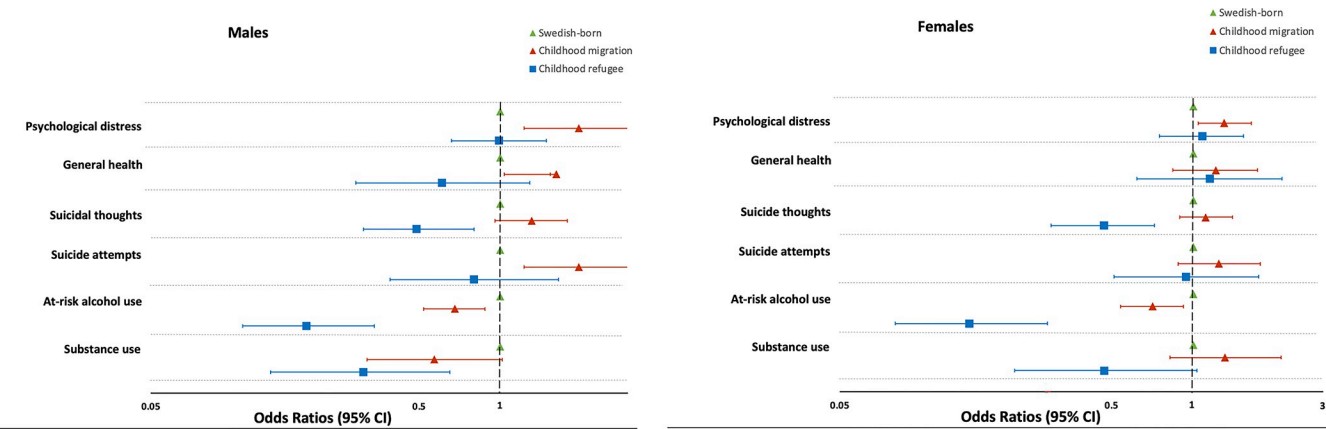

**Fig 1. Odds ratios for health and risk behaviours based on childhood refugee experience and migrant childhood experience in 39,608 males and 49,808 females aged 18–64.**

Overall, females reported slightly lower proportions of at-risk alcohol use, but the association was similar in strength between the sexes in the logistic regression adjusted for confounders (refugee males aOR 0.19, 0.11–0.34 and refugee females aOR 0.15, 0.08–0.29) (Table 3 and Fig 1). We saw a similar association in the adjusted analyses concerning substance use, but it was only statistically significant in refugee males aOR 0.31 (0.14–0.65).

### Differences by age

In an additional analysis, we stratified by age-groups (Table 4).

In age-group startified analysis, we found that the participant's age also reflects the time lived in Sweden, with younger adults (18–25 years old) having lived in Sweden for 1–24 years, and older adults (26–64 years) for 9 to 47 years.

### Association between refugee or migrant experience in childhood and self-reported mental and general health stratified by age

Among young adults, being a refugee in childhood was associated with lower odds of suicide thoughts (aOR 0.47, 0.29–0.75), and better self-rated general health (aOR 0.35, 0.19–0.64) than Swedish-born. There was no association between being a migrant in childhood and any of the outcomes in young adulthood. Among adults, being a refugee was associated with psychological distress (aOR 1.71, 1.18–2.48) Further, being a migrant in childhood was associated with psychological distress (aOR 1.39, 1.12–1.73), suicide thoughts (aOR 1.31, 1.06–1.63), and worse general health (aOR 1.44, 1.07–1.93), compared to Swedish-born adults (Table 4).

### The association between refugee or migrant experience in childhood and self-reported alcohol and substance stratified by age

Amongst young adults, those with refugee experience had lower odds for substance use (aOR 0.24, 0.10–0.57). Among adults, being a refugee was associated with lower odds of at-risk alcohol use (aOR 0.27, 0.16–0.47) than Swedish-born peers.

## Discussion

In this national population-based study, we found that, in general, being a refugee in childhood was *not* associated with worse self-rated poor health in adulthood compared to those with a migrant experience in childhood and Swedish-born peers when adjusted for age and sexual- and gender minority status. Individuals with a refugee or migrant experience in childhood had a lower risk of at-risk alcohol and substance use than Swedish-born. However, one exception was that adults aged 26–64 years, with a refugee or migrant experience in childhood

**Table 4. Associations between childhood refugee experience and migrant childhood experience and risk behaviours in 89,416 individuals aged 18–64 from the Swedish National Public Health Survey.** Estimates are from multivariable logistics regression models (models adjusted for sexual minority status). Stratified by age.

| | Psychological distress | Suicide thoughts | Suicide attempts | At-risk alcohol use | Substance use | General health |
|---|---|---|---|---|---|---|
| Young adults | | | | | | |
| Swedish-born | Ref (1) | Ref (1) | | Ref (1) | Ref (1) | |
| Refugee experience in childhood | 0.90 (0.60–1.34) | **0.47 (0.29–0.75)** | 1.02 (0.56–1.87) | **0.12 (0.06–0.24)** | **0.24 (0.10–0.57)** | **0.35 (0.19–0.64)** |
| Migrant experience in childhood | 1.05 (0.73–1.51) | 1.05 (0.73–1.51) | 1.24 (0.73–2.12) | **0.36 (0.24–0.54)** | 0.88 (0.53–1.46) | 1.14 (0.53–2.43) |
| Adults | | | | | | |
| Swedish-born | Ref (1) | Ref (1) | Ref (1) | Ref (1) | Ref (1) | Ref (1) |
| Refugee experience in childhood | **1.71 (1.18–2.48)** | 0.77 (0.49–1.20) | 0.81 (0.39–1.66) | **0.27 (0.16–0.47)** | 1.50 (0.75–2.97) | 1.21 (0.67–2.18) |
| Migrant experience in childhood | **1.39 (1.12–1.73)** | **1.31 (1.06–1.63)** | **1.71 (1.22–2.39)** | 0.84 (0.68–1.03) | 0.84 (0.51–1.38) | **1.44 (1.07–1.93)** |

had higher odds of psychological distress compared to Swedish-born peers. There were no apparent sex differences in any of the associations we examined. One explanation for our finding of no association between refugee experience in childhood and subsequent health could be Sweden's historically inclusive integration system, which provides access to healthcare, education for all children, and a well-developed social security system Although Sweden has implemented more restrictive integration policies since 2015, most of the participants arrived before this shift in policy, and thus experiences the older more inclusive system. Therefore, these results may be attributed, in part, to factors related to the immigration system. However, it's important to recognize that the inclusivity of the Swedish system is relative. The Swedish system, like any integration system, faces its own challenges. Assessing the impact of these challenges on refugees and migrants in this study is however impossible, given that the participants migrated to Sweden over a long period of time. The nature of these challenges is most likely depending on the timing of one's immigration to Sweden and the diverse experiences that different populations bring with them.

This is the first study, using self-assessment, examining if refugee experience in childhood is a risk factor for later negative health consequences in a nationally representative sample, including individuals with refugee and migrant backgrounds analyzed separately. Our findings show that individuals who were refugees and migrants as children were more likely to report poorer mental health (psychological distress, suicidal ideation, and suicide attempts) in adulthood. This is, at least partially, explained by factors like sexual- and gender identity and age since there is no association when adjusting for these factors. The number of individuals identifying as SGM was higher in refugees and migrants compared to Swedish-born peers probably due to the fact that being SGM is a valid reason for seeking asylum. However, many arrived at a young age, which mean that they probably did not present their own individual reason for asylum, rather their parents did. Another possible explanation is that these two groups were younger than the Swedish-born group, and that identification as sexual- or gender-minority is more prevalent among younger than older people. When we stratified by age group, individuals 26–64 years of age with experience of refugee experience in childhood had higher odds for psychological distress than the Swedish-born.

Our findings on at-risk alcohol- and substance use align with earlier research showing lower rates of self-assessed substance and at-risk alcohol use disorder in both migrant and refugee groups in Sweden [38]. Earlier studies have included refugees and migrants, regardless of age at migration. However, they do not align with earlier research on health-care utilization showing an association between former refugee status and later mental ill-health [8] nor earlier Swedish research studying health-care utilization for substance use. Our findings also differ from earlier research on the prolonged impact of childhood adversity [39]. A potential explanation is that individuals who migrate in childhood, regardless of reason, have increased access to resources enabling better social integration as than those who migrated in adulthood. For example, being a refugee in childhood provides increased opportunities to learn Swedish and better access to resources contributing to cultural integration [13]. Another possible explanation is that the sample included in this study is different from most existing studies on childhood refugees. Earlier studies have been restricted to specific nationalities and are rarely nationally representative. This could be relevant since this study captures different experiences and time periods. Further, it could indicate that the studies using registers with health care data investigate a different population. Most of these have looked at healthcare utilization, assessment by mental health professionals/clinicians, other populations, and in-patient care [16,18,40,41]. The studies investigating healthcare utilization should be considered complementary to studies based on self-assessment. Healthcare utilization studies consider only those in contact with healthcare or receiving services, while the present study based on self-

assessment reflects a nationally representative sample, regardless of healthcare needs or access to services.

Nevertheless, this could also indicate that being a refugee in childhood and arriving in a high-income country such as Sweden should not be considered a risk factor for poorer health in adulthood. Earlier studies showing this may instead reflect that refugees and migrants, in general, are younger and have higher proportions of sexual- or gender minority status, alone or in combination with refugee or migration experience. This needs to be confirmed by further studies, preferably including a diverse set of nationalities, across differing reasons for asylum applications, and using a longitudinal design.

## Strengths and limitations

A major strength is that the results of this large and nationally representative study can be generalizable to the Swedish adult population. Further, missing data was very low, especially for a national survey of this size. Another strength is the use of population weights in the analysis, which partially compensates for non-response [28]. A further benefit of the present study is that it could mimic a long-term follow-up of the participants since their migration date occurred before the ascertainment of the outcomes. The average time between arrival in Sweden and follow-up was 16 years for refugees and 34.71 years for migrants. This gave us the unique possibility to measure the impact of being a childhood refugee or migrant on outcomes in adulthood. The large sample size also allowed us to disaggregate the data to compare different age groups, sex, and refugee/migrant status. Further, it included refugees and migrants from diverse countries/regions (Table C in S1 Text and Fig 1), including Syria, Afghanistan, Iraq, Iran, former Yugoslavia, and Somalia.

Some considerations to note include a change in the instrument used to assess psychological distress (from Kessler-6 to GHQ-5) between 2018 and 2020. First, there are some limitations in general when using brief questionnaires such as K6 and GHQ-5 instead of diagnostic interviews. Ideally, distress would have been assessed using multiple time points and with a diagnostic interview to provide a more detailed understanding of the presence of mental disorders. However, this was not possible in this study but future studies using these methods are needed. Second, these instruments have different recall periods (two versus four weeks) which might influence the findings. Third, there are studies that show that Kessler-6 outperforms GHQ-12 in capturing symptoms of psychological distress [42]. We addressed this by analysing the data separately for 2018 and 2020, but we found similar results. The prevalence of psychological distress in our study was generally lower compared to previous studies conducted in the USA and Canada [43,44], but higher than reported in a Finnish study [45]. However, it is important to note that these numbers are not directly comparable due to several factors. Firstly, our study excluded certain groups, such as individuals over 65 years of age, which may have influenced the prevalence rates. Secondly, different instruments were used to assess psychological distress in the studies, making direct comparisons challenging. Further, while the categorization of continuous variables results in a loss of statistical power, but it also facilitates comparison with similar studies [46]. Also, even though this study has a comparably large sample size, we still had relatively few individuals in some age groups and outcomes (see Table 2). Optimally, we would have stratified by country of origin, but unfortunately, the number of refugee individuals was too small to allow for such analyses.

Another limitation could be the use of self-reported questionnaires, as they may not fully capture a spectrum of mental health and mental disorders. Previous research has argued the opposite, that self-report questionnaires are too inclusive [7]. Another problem with the use of self-report questionnaires might also be that they are validated mainly using Western/

European origin populations. Studies are needed to ensure the validity of these instruments for this population. The surveys were only distributed in three languages. However, all included in the study arrived before 18 years of age and hence would have been included in the Swedish school system, and/or been entitled to Swedish language education that is offered at no cost to immigrants.

Some inherent challenges lie in creating the categories of refugees and migrants, since these groups are not fixed in real life. For example, one consideration may be that someone might have fled at the beginning of a conflict, but might be labeled a migrant in the present data. On the other hand, someone might have migrated from a war-torn country like Syria for job prospects, but would be categorized as a refugee. Lastly, using arrival before age 18 as an indicator of exposure in childhood is novel, but it also has the inherent problem that childhood is an extended period, and exposures at different time points are likely to affect children differently. For example, a child arriving close after birth will have a very different experience than someone arriving at 17 years of age both in the timing of exposure, the amount of exposure to potential adversities, and possibilities to integrate into society. This has been shown in earlier studies but was not possible to investigate here, and we recommend further studies on this.

In summary, we found that those with refugee or migrant experience in childhood generally did not have worse self-rated general health or mental health in adulthood compared to those born in Sweden when adjusting for confounders. This suggests that we revisit the assumption that refugee experience in childhood is a risk factor for later mental and physical ill-health. Based on present findings, policies should not reflect an assumption that refugee populations are at higher risk for health problems in the long term. In order to promote better health for those in need, it is critical to prioritize resources for younger refugees and those with double minority status. Through this shift in focus, we can more effectively address disparities in health and ensure that support is provided where it is most needed.

## Supporting information

**S1 Checklist. STROBE statement—checklist of items that should be included in reports of *cross-sectional studies*.**
(DOCX)

**S1 Text.** Table A in S1 Text: A detailed description of the variables in the Swedish National Public Health Survey 'Health on equal terms' (2018–20) and the variables that we analysed in this study. Table B in S1 Text: How we identified migrant and refugee experience in childhood among participants in the National Public Health Survey in 2018 and 2020. The strategy outlined was developed in consultation with Swedish Board of Migration. Table C in S1 Text: Proportions of country of origin, sex and time in Sweden among those with a refugee experience in childhood, those with a refugee experience in childhood, or Swedish-born. Stratified by age groups. The numbers are unweighted counts (n) and unweighted proportions (%). Table D in S1 Text. Development of the model. Stratified by sex. Table E in S1 Text. Development of the model. Stratified by sex. Fig A S1 in Text: Illustration of geographical origin (unweighted), time of conflicts, and age groups[1].
(DOCX)

## Acknowledgments

The data used in this study were from the Swedish National Public Health Survey in 2018 and 2020 conducted by the Swedish National Public Health Institute and Statistics Sweden.

## Author Contributions

**Conceptualization:** Erica Mattelin, Frida Fröberg.

**Data curation:** Erica Mattelin, Frida Fröberg.

**Formal analysis:** Erica Mattelin, Frida Fröberg.

**Investigation:** Erica Mattelin, Frida Fröberg.

**Methodology:** Frida Fröberg.

**Supervision:** Amal R. Khanolkar, Laura Korhonen, Frida Fröberg.

**Writing – original draft:** Erica Mattelin.

**Writing – review & editing:** Erica Mattelin, Amal R. Khanolkar, Laura Korhonen, Jill. W. Åhs, Frida Fröberg.

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
