## [Decision Letter · Decision Letter 0]

30 May 2023

PGPH-D-23-00525

Is refugee experience in childhood a risk for poorer health in adulthood - A Swedish national survey study

Dear Dr. Mattelin,

Thank you for submitting your manuscript to PLOS Global Public Health. After careful consideration, we feel that it has merit but does not fully meet PLOS Global Public Health’s publication criteria as it currently stands. Therefore, we invite you to submit a revised version of the manuscript that addresses the points raised during the review process.

We look forward to receiving your revised manuscript.

Kind regards,

Godfred Boateng

Academic Editor

Journal Requirements:

1. Our staff editors have determined that your manuscript is likely within the scope of our Global Mental Health: challenges, opportunities, and the future of the field. This editorial initiative is headed by a team of Guest Editors for PLOS GPH: Rochelle Burgess (University College of London) and Dixon Chibanda (University of Zimbabwe and London School of Tropical Medicine and Hygiene). The Collection invites researchers to submit original research which engages with, or disrupts, the urgent needs across the global mental health landscape. We especially encourage submissions of studies that critically interrogate the status quo of the field and that involve inter-/trans-disciplinary approaches and those which share perspectives from underrepresented global regions and communities.

 Additional information can be found on our announcement page: https://collections.plos.org/call-for-papers/global-mental-health-opportunities-challenges/ 

If you would like your manuscript to be considered for this collection, please let us know in your cover letter and we will ensure that your paper is treated as if you were responding to this call.  Please note that being considered for the Collection does not require additional peer review beyond the journal’s standard process and will not delay the publication of your manuscript if it is accepted by PLOS GPH. If you would prefer to remove your manuscript from collection consideration, please specify this in the cover letter.

2. Your manuscript is missing the following sections: Introduction. Please ensure these are present, and in the correct order, and that any references to subheadings in your main text are correct. An outline of the required sections can be consulted in our submission guidelines here:

https://journals.plos.org/globalpublichealth/s/submission-guidelines#loc-parts-of-a-submission

Additional Editor Comments (if provided):

Reviewers' comments:

Reviewer's Responses to Questions

**Comments to the Author**

1. Does this manuscript meet PLOS Global Public Health’s publication criteria? Is the manuscript technically sound, and do the data support the conclusions? The manuscript must describe methodologically and ethically rigorous research with conclusions that are appropriately drawn based on the data presented.

Reviewer #1: No

Reviewer #2: Yes

Reviewer #3: Yes

2. Has the statistical analysis been performed appropriately and rigorously?

Reviewer #1: No

Reviewer #2: Yes

Reviewer #3: No

3. Have the authors made all data underlying the findings in their manuscript fully available (please refer to the Data Availability Statement at the start of the manuscript PDF file)?

Reviewer #1: Yes

Reviewer #2: No

Reviewer #3: Yes

4. Is the manuscript presented in an intelligible fashion and written in standard English?

Reviewer #1: Yes

Reviewer #2: Yes

Reviewer #3: Yes

5. Review Comments to the Author

Reviewer #1: Please consider confounding by country and religion of origin for refugees. Alcohol use may be negatively associated with being muslim. Are the country of origin of migrants and refugees similar? Do the have differing distributions from different regions around the world that may account for differences in things like substance abuse, psychological distress? When you refer to multivariable regression are you referring to multinomial logistic regression? That is, are all of the groups in the same logistic regression model? Also, are there any additional confounders that you can adjust for in those models? Among the refugee women is it possible that they are unlikely to report suicidal thoughts? They appear to have a higher odds of suicidal attempts, but lower for suicidal thoughts. Could this be related to differences in how the questions are interpreted or comfort and reporting suicidal thoughts? Overall it could be that it cultural and religious status account for some of the differences in the self-reported metrics as opposed to immigrant or refugee status per say. It would help to differentiate that in the paper and consider sources of confounding.

Reviewer #2: Is refugee experience in childhood a risk for poorer health in adulthood - A Swedish

national survey study – PGPH-D-23-00525

The authors examined whether adults 18-64 years with a childhood refugee experience were more likely to self-rate their mental/general health as poor compared to non-refugee migrants and Swedish-born.

The manuscript presents rigorous original research findings that contribute to refugee studies and especially the long-term relationships between being a refugee and adult health/wellbeing. The authors performed technically sound inferential statistical analysis and described and discussed the findings in sufficient detail. Overall, the paper is well organized and clearly written. The data appears to support the conclusions drawn by the authors. However, the authors may focus on a few areas below to strengthen the manuscript.

Background

Your background is too short (less than 2 pages). There is much current literature regarding childhood refugee experiences and the risk of poor health in adulthood from a Swedish perspective. Kindly consider adding more detail to the background to provide a more compelling argument in which you would situate the need for your study.

Methods and Materials

In your paper, it is not clear how you accounted for missing data. Kindly consider giving more details.

Refugees’ and immigrants' ethnicity and socioeconomic status are key in determining health and how they cope. I was expecting to see other key demographic variables such as participants’ employment status in Table 1. Kindly consider.

The Kessler Psychological Distress Scale measures a person’s psychological distress/emotional state within a period of 4 weeks. But you were seeking to measure the mental health of adults who migrated as refugees at a younger age. I wonder whether measuring the emotional state of your participants in the past 4 weeks only properly captured the psychological distress of the participants. The participants’ mental health might have been a cumulative result of events linked to being a refugee at a young age (which goes beyond 4 weeks). Why did the authors measure social distress only within the past 4 weeks and not beyond, such as the past 6 months, 12 months, etc.? Could this short period influence your results in any way? If so, could it be a limitation to how far you could interpret your findings? Kindly consider.

Discussion

Page 22 paragraph 3 line 2: ‘aren’t’, should be written in full as ‘are not’

Statements and Declarations

Page 23 paragraph 1 line 8: Check the sentence and change ‘However’ to ‘however’ appropriately.

Page 24 under ‘Availability of data and materials’, check the statement: ‘ However, data from the The Swedish National…’ and remove ‘The’ appropriately.

Reviewer #3: Comments to the Author(s)

This manuscript examined whether adults 18-64 years with a childhood refugee experience were more likely to self-rate their mental or general health as poor, or have risk behaviors, compared to non-refugee migrants and Swedish-born using a national representative sample. While I commend the authors for a well-written paper, there are several issues that need to be addressed to achieve what the article aims to do, along with clarifications of the methods. My comments are shown below.

Major comments

Introduction and justification of the study

I believe a brief section on Swedish policies and programs for refugees and migrants would help contextualize the study findings and help readers appreciate the findings. For instance, such a section could brief describe the Swedish immigrants’ profile and current and historical policies on immigration.

Methodological and Analytical issues

Even though I appreciate the effort and creativity of the author(s) in creating their variable, I wonder if “Those with refugee experience in childhood (0-17 years)” adequately captures the refugee experience. Would older children better capture the refugee experience, say 10-17 than 0-17? I am particularly concerned that 0-5year olds may not remember the refugee experience and may have similar experience to those who migrated in childhood for reasons not associated with refugee status or asylum seeking.

Further, it would have love to see the authors extend the control variables to include socio-economic variables such as household income, education, religion etc. if these are available in the Swedish National Public Health Survey.

Even though the authors acknowledged and addressed the change in instruments used to assess psychological distress, I wonder how the differences in recall period between the previous two weeks of the GHQ-5 and the past four weeks with (Kessler-6) might affect the results.

Results

I suggest the authors provide the distribution of their outcome variables and briefly indicate what percentage for instance were psychological distressed, and how did that vary by status in Sweden. How did the authors results on psychological distress compare with other countries that used a similar instrument. The supplementary file did not seem to contain the distribution of the outcome variables.

On page, 17, the authors indicated that among adults, being a refugee was associated with lower odds of at-risk alcohol use than Swedish-born. Could this be of a result of not controlling for or accounting for the religious and cultural affiliations of immigrants?

Discussion and conclusion

I believe the authors could strengthen the discussion by indicating the potential policy implications of their study for Sweden and other high-income countries that accept migrants.

Minor

The authors indicated on page 22 that the surveys were only distributed in three languages, did this limit potential participation?

6. PLOS authors have the option to publish the peer review history of their article (what does this mean?). If published, this will include your full peer review and any attached files.

**Do you want your identity to be public for this peer review?** For information about this choice, including consent withdrawal, please see our Privacy Policy.

Reviewer #1: No

Reviewer #2: No

Reviewer #3: No

---

## [Decision Letter · Decision Letter 1]

31 Aug 2023

PGPH-D-23-00525R1

Is refugee experience in childhood a risk for poorer health in adulthood? - A Swedish national survey study

Dear Dr. Mattelin,

Thank you for submitting your manuscript to PLOS Global Public Health. After careful consideration, we feel that it has merit but does not fully meet PLOS Global Public Health’s publication criteria as it currently stands. Therefore, we invite you to submit a revised version of the manuscript that addresses the points raised during the review process.

We look forward to receiving your revised manuscript.

Kind regards,

Godfred Boateng

Academic Editor

Journal Requirements:

1.Our staff editors have determined that your manuscript is likely within the scope of our Global Mental Health: challenges, opportunities, and the future of the field. This editorial initiative is headed by a team of Guest Editors for PLOS GPH: Rochelle Burgess (University College of London) and Dixon Chibanda (University of Zimbabwe and London School of Tropical Medicine and Hygiene). The Collection invites researchers to submit original research which engages with, or disrupts, the urgent needs across the global mental health landscape. We especially encourage submissions of studies that critically interrogate the status quo of the field and that involve inter-/trans-disciplinary approaches and those which share perspectives from underrepresented global regions and communities.

 Additional information can be found on our announcement page: https://collections.plos.org/call-for-papers/global-mental-health-opportunities-challenges/ 

If you would like your manuscript to be considered for this collection, please let us know in your cover letter and we will ensure that your paper is treated as if you were responding to this call.  Please note that being considered for the Collection does not require additional peer review beyond the journal’s standard process and will not delay the publication of your manuscript if it is accepted by PLOS GPH. If you would prefer to remove your manuscript from collection consideration, please specify this in the cover letter."

Reviewers' comments:

Reviewer's Responses to Questions

**Comments to the Author**

1. If the authors have adequately addressed your comments raised in a previous round of review and you feel that this manuscript is now acceptable for publication, you may indicate that here to bypass the “Comments to the Author” section, enter your conflict of interest statement in the “Confidential to Editor” section, and submit your "Accept" recommendation.

Reviewer #2: All comments have been addressed

Reviewer #3: All comments have been addressed

2. Does this manuscript meet PLOS Global Public Health’s publication criteria? Is the manuscript technically sound, and do the data support the conclusions? The manuscript must describe methodologically and ethically rigorous research with conclusions that are appropriately drawn based on the data presented.

Reviewer #2: Yes

Reviewer #3: Yes

3. Has the statistical analysis been performed appropriately and rigorously?

Reviewer #2: Yes

Reviewer #3: Yes

4. Have the authors made all data underlying the findings in their manuscript fully available (please refer to the Data Availability Statement at the start of the manuscript PDF file)?

Reviewer #2: No

Reviewer #3: Yes

5. Is the manuscript presented in an intelligible fashion and written in standard English?

Reviewer #2: Yes

Reviewer #3: Yes

6. Review Comments to the Author

Reviewer #2: (No Response)

Reviewer #3: Comments to the authors

I commend the authors for addressing most of my comments and suggestions satisfactorily. However, there are a few minor comments on the updated manuscript. These comments are shown below:

The authors on page 4 indicated that “Many refugee children also experience mental health issues during the first few years after arrival in a host country. Refugee children have a higher prevalence of PTSD, depression, and anxiety disorders shortly after resettlement in a host country than native-born peers (7). In general, a childhood refugee experience was not associated with worse self-rated health or risk behaviours in adulthood when considering age and sexual- and gender-minority status”. It might be useful to provide a broader context and indicate the systemic factors in Sweden that often lead to these experiences.

It might also be important to contextualize the key finding of the study that “In general, a childhood refugee experience was not associated with worse self-rated health or risk behaviours in adulthood when considering age and sexual- and gender-minority status”. This might be due to the various protective programs offered by Swedish immigration policy.

In the first paragraph of the discussion section, it might be useful for the authors to provide some context and provide potential reasons for their findings. The authors attempted doing this at the later part of the strengths and limitations section on page 23. It might be relevant to move those parts to after the first paragraph of the discussion section.

There a few typographical errors and the manuscript might benefit from another round of review. For instance, on in the abstract lines 54 and 55, it reads: were analysed using logistic regression, adjusted for age and sexual- and gender-minority (SGM) identity, and run stratified by sex”

7. PLOS authors have the option to publish the peer review history of their article (what does this mean?). If published, this will include your full peer review and any attached files.

**Do you want your identity to be public for this peer review?** For information about this choice, including consent withdrawal, please see our Privacy Policy.

Reviewer #2: No

Reviewer #3: **Yes: **Joseph Kangmennaang

---

## [Editor Report · Decision Letter 2]

17 Oct 2023

Is refugee experience in childhood a risk for poorer health in adulthood? - A Swedish national survey study

PGPH-D-23-00525R2

Dear Miss Mattelin,

We are pleased to inform you that your manuscript 'Is refugee experience in childhood a risk for poorer health in adulthood? - A Swedish national survey study' has been provisionally accepted for publication in PLOS Global Public Health.

Best regards,

Godfred Boateng

Academic Editor